# The learnability of Pauli noise

**Senrui Chen** [1,5] ✉, **Yunchao Liu** [2,5] ✉, **Matthew Otten** [3], **Alireza Seif**[1], **Bill Fefferman** [4] & **Liang Jiang** [1] ✉

Recently, several quantum benchmarking algorithms have been developed to characterize noisy quantum gates on today's quantum devices. A fundamental issue in benchmarking is that not everything about quantum noise is learnable due to the existence of gauge freedom, leaving open the question what information is learnable and what is not, which is unclear even for a single CNOT gate. Here we give a precise characterization of the learnability of Pauli noise channels attached to Clifford gates using graph theoretical tools. Our results reveal the optimality of cycle benchmarking in the sense that it can extract all learnable information about Pauli noise. We experimentally demonstrate noise characterization of IBM's CNOT gate up to 2 unlearnable degrees of freedom, for which we obtain bounds using physical constraints. In addition, we show that an attempt to extract unlearnable information by ignoring state preparation noise yields unphysical estimates, which is used to lower bound the state preparation noise.

Characterizing quantum noise is an essential step in the development of quantum hardware[1,2]. Remarkably, despite recent progress in both gate-level and scalable noise characterization methods[3–16], the full characterization of the noise channel of a single CNOT/CZ gate remains infeasible. This is unlikely to be caused by limitations of existing benchmarking algorithms. Instead, it is believed to be related to the fundamental question of what information about a quantum system can be learned, in a setting where initial states, gates, and measurements are all subject to unknown quantum noise. It is well-known that some information about quantum noise can be learned (such as the gate fidelity learned by randomized benchmarking[3–7] or cycle benchmarking[9]), but not everything can be learned (due to the gauge freedom in gate set tomography[17–19]). The boundary of learnability of quantum noise – a precise understanding of what information is learnable and what is not, still remains an open question.

Recently, there has been an interest in formulating noise characterization as learning unknown gate-dependent Pauli noise channels[9,11]. This is motivated by randomized compiling, a technique that has been proposed to suppress coherent errors via inserting random Pauli gates[20,21]. As an added benefit, randomized compiling twirls the gate-dependent CPTP noise channel into Pauli noise, thus reducing the number of parameters to be learned. Note that the

twirled Pauli noise channel corresponds to the diagonal of the process matrix of the CPTP map, so Pauli noise learning is a necessary step for characterizing the CPTP map, regardless of whether randomized compiling is performed.

However, even under this simplified setting of Pauli noise learning, all prior experimental attempts can only partially characterize the noise channel of a single CNOT/CZ gate[21–23], which only has 15 degrees of freedom. A natural question is whether this limitation is caused by the fundamental unlearnability of the noise channel, and if so, which part of the noise channel and how many degrees of freedom among the 15 are unlearnable?

In this paper, we give a precise characterization of what information in the Pauli noise channel attached to Clifford gates is learnable, in a way that is robust against state preparation and measurement (SPAM) noise. We develop a systematic method for characterizing learnable degrees of freedom of a Clifford gate set using notions from algebraic graph theory and show that learnable information exactly corresponds to the cycle space of the Pauli pattern transfer graph, while unlearnable information exactly corresponds to the cut space. This characterization can be used to write down a list of linear functions of the noise model that corresponds to all independent learnable degrees of freedom. As an example, we show that the Pauli noise

[1]Pritzker School of Molecular Engineering, University of Chicago, Chicago, IL 60637, USA. [2]Department of Electrical Engineering and Computer Sciences, University of California, Berkeley, CA 94720, USA. [3]HRL Laboratories, LLC, 3011 Malibu Canyon Rd., Malibu, CA 90265, USA. [4]Department of Computer Science, University of Chicago, Chicago, IL 60637, USA. [5]These authors contributed equally: Senrui Chen, Yunchao Liu. ✉e-mail: csenrui@uchicago.edu; yunchaoliu@berkeley.edu; liang.jiang@uchicago.edu

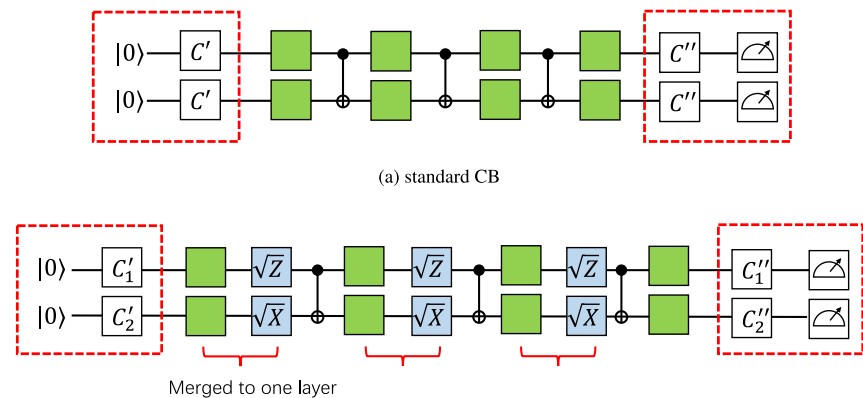

**Fig. 1 | Cycle benchmarking for learning the Pauli noise channel of a CNOT gate.**
**a** Standard CB circuits, where CNOT gates are interleaved by random Pauli gates (green boxes), with initial stabilizer states and Pauli basis measurements (red boxes). **b** CB circuits with additional interleaved single qubit Clifford gates (blue boxes).

channel of an arbitrary 2-qubit Clifford gate has at most 2 unlearnable degrees of freedom. We perform an experimental characterization of a CNOT gate on IBM Quantum hardware[24] up to 2 unlearnable degrees of freedom. Although the unlearnable information cannot be estimated with high precision, we can determine a feasible region of those freedoms using the constraint that the noise model must be physical (i.e., all Pauli error rates are nonnegative).

A corollary of our result is that cycle benchmarking is optimal in the setting we consider, in the sense that it can learn all the information that is learnable. This reveals a fundamental fact about noise benchmarking, namely that cycle benchmarking – the idea of repeatedly applying the same gate sequence interleaved by single qubit gates, is the "right" algorithm for benchmarking Clifford gates, because of the fact that learnable information forms a cycle space. As an interesting side remark, the term "cycle" in cycle benchmarking originally refers to parallel gates applied in a clock cycle. Here we show that the term can also be understood in a graph-theoretical context.

In addition, we also explore ways to overcome the unlearnability barrier. It has been recognized that the unlearnability does not apply if the initial state $|0\rangle^{\otimes n}$ can be prepared perfectly[15,23], and it has been suggested that state preparation noise could be much smaller than gate and/or measurement noise in practice[25–27], which would make gate noise fully learnable up to small error. We develop an algorithm based on cycle benchmarking that fully learns gate-dependent Pauli noise channel assuming perfect initial state preparation, and experimentally demonstrate the method on IBM's CNOT gate. Based on the experiment data, we conclude that this assumption is unlikely to be correct in our experiment as it gives unphysical estimates that are outside the feasible region we determined. Furthermore, we use the data to obtain a lower bound on the state preparation noise and conclude that it has the same order of magnitude as gate noise on the device we used. Therefore, the issue of unlearnability is a practically relevant concern, for which the noise on initial states is an important factor that cannot be neglected on current quantum hardware.

## Results
### Theory of learnability
We start by considering the learnability of the Pauli noise channel of a single $n$-qubit Clifford gate. A Pauli channel can be written as

$$\Lambda(\cdot) = \sum_{a \in \mathsf{P}^n} p_a P_a(\cdot) P_a, \tag{1}$$

where $\{p_a\}$ is a probability distribution on $\mathsf{P}^n = \{I, X, Y, Z\}^n$. The goal is to learn this distribution, which has $4^n - 1$ degrees of freedom. Considering $\Lambda$ as a linear map, its eigenvectors exactly correspond to all $n$-qubit Pauli operators, as

$$\Lambda(P_a) = \lambda_a P_a, \quad \forall a \in \mathsf{P}^n \tag{2}$$

where $\lambda_a = \sum_{b \in \mathsf{P}^n} p_b(-1)^{\langle a,b \rangle}$ is the Pauli fidelity associated with the Pauli operator $P_a$. Therefore $\Lambda$ is a linear map with known eigenvectors and unknown eigenvalues, so a natural way to learn $\Lambda$ is to first learn all the Pauli fidelities $\lambda_a$, and then reconstruct the Pauli errors via $p_a = \frac{1}{4^n} \sum_{b \in \mathsf{P}^n} \lambda_b(-1)^{\langle a,b \rangle}$.

The convenience of working with Pauli fidelities is further demonstrated by the fact that some Pauli fidelities can be directly learned by cycle benchmarking, even with noisy state preparation and measurement. For example, consider the CNOT gate which maps the Pauli operator $IX$ to itself. Figure 1(a) shows the cycle benchmarking circuit. Imagine that we put the Pauli operator $IX$ after the left red box and evolve it with the circuit, then the evolved operator (before the right red box) equals $\lambda_{IX}^3 \cdot IX$, up to a $\pm$ sign (which comes from the random Pauli gates and can always be accounted for during post-processing). Here we use the convention that the noise channel happens before each CNOT gate. In experiments, we prepare a $+1$ eigenstate of $IX$ (such as $|+\rangle|+\rangle$), measure the expectation value of $IX$ at the end, and average over random Pauli twirling sequences. These SPAM operations are noisy and are represented as the red boxes. It is shown[9] that the measured expectation value equals

$$\mathbb{E}\langle IX \rangle = A_{IX} \cdot \lambda_{IX}^d \tag{3}$$

where the expectation is over random Pauli twirling gates and randomness of quantum measurement, and $A_{IX}$ depends on SPAM noise but is independent of circuit depth $d$. From this $\lambda_{IX}$ can be learned by estimating the observable $IX$ at several different depths and perform a curve fitting.

The Pauli operator $IX$ is special as it is invariant under CNOT. Consider another example: CNOT maps $XZ$ to $YY$ and vice versa. Consider Fig. 1(b) where we insert additional layers of single-qubit Clifford gates $\sqrt{Z} \otimes \sqrt{X}$ that also maps $XZ$ to $YY$ and vice versa (up to a minus sign that can always be accounted for during post-processing). After $XZ$ picks up a coefficient $\lambda_{XZ}$ in front of the CNOT gate, it gets mapped to $\lambda_{XZ} \cdot YY$ by CNOT but then rotated back to $\lambda_{XZ} \cdot XZ$ by $\sqrt{Z} \otimes \sqrt{X}$. Following the same argument we conclude that both $\lambda_{XZ}$ and $\lambda_{YY}$ are

learnable. For simplicity here we make an assumption that single qubit gates are noiseless, motivated by the fact that single qubit gates are 1-2 magnitudes less noisy than 2-qubit gates on today's quantum hardware[24]. In practice, it is a standard assumption to model noise on single-qubit gates as gate-independent (e.g.[23]), and our noise characterization result can be interpreted as the noise channel induced by a dressed cycle which consists of a CNOT gate and two single-qubit gates[20].

The main challenge comes with the next example: CNOT maps $IZ$ to $ZZ$ and vice versa. By directly applying cycle benchmarking as in Fig. 1(a) (with even depth $d$) we obtain

$$\mathbb{E}\langle IZ \rangle = A_{IZ} \cdot \lambda_{IZ}\lambda_{ZZ}\lambda_{IZ}\lambda_{ZZ}\cdots = A_{IZ}\left(\lambda_{IZ}\lambda_{ZZ}\right)^{d/2}, \qquad (4)$$

and curve fitting gives $\sqrt{\lambda_{IZ}\lambda_{ZZ}}$ (similar results have been obtained in[9,21–23]). To learn $\lambda_{IZ}$, we may consider applying the same technique in Fig. 1(b). However, the problem is that once $IZ$ gets mapped to $ZZ$, it cannot be rotated back to $IZ$ because $I$ is invariant under single qubit unitary gates. The main difference between this example and previous examples is that here the Pauli weight pattern (an $n$-bit binary string with 0 indicating identity and 1 indicating non-identity) changes from 01 to 11, thus making the single qubit rotation tool inapplicable.

In fact we can go on to prove that $\lambda_{IZ}$ (as well as $\lambda_{ZZ}$) is unlearnable. Here unlearnable means that there exists two noise models such that the parameter $\lambda_{IZ}$ is different, but the two noise models are indistinguishable by any quantum experiment, meaning that any quantum experiment generates exactly the same output statistics with the two noise models. The result also generalizes to arbitrary $n$-qubit Clifford gates.

**Theorem 1.** Given an $n$-qubit Clifford gate $\mathcal{G}$ and an $n$-qubit Pauli operator $P_a$, the Pauli fidelity $\lambda_a$ of the noise channel attached to $\mathcal{G}$ is learnable if and only if $\text{pt}(\mathcal{G}(P_a)) = \text{pt}(P_a)$. Here pt denotes the Pauli weight pattern.

The "if" part follows directly from cycle benchmarking as discussed above. For the "only if" part, when $\text{pt}(\mathcal{G}(P_a)) \neq \text{pt}(P_a)$, we construct a gauge transformation to prove the unlearnability of $\lambda_a$, following ideas from gate set tomography[17–19]. A gauge transformation is an invertible linear map $\mathcal{M}$ that converts a noise model (initial states $\rho_i$, POVM operators $E_j$, noisy gates $G_k$) to a new noise model as

$$\rho_i \mapsto \mathcal{M}(\rho_i), \quad E_j \mapsto (\mathcal{M}^{-1})^\dagger(E_j), \quad G_k \mapsto \mathcal{M} \circ G_k \circ \mathcal{M}^{-1}, \qquad (5)$$

with the constraint that the new noise model is physical. Note that the old and new noise models are indistinguishable by definition. To construct such a gauge transformation, as $\text{pt}(\mathcal{G}(P_a)) \neq \text{pt}(P_a)$, there exists a bit on which the two Pauli weight patterns differ. We then define $\mathcal{M}$ as a single-qubit depolarizing noise channel on the corresponding qubit. In this way we can show that the old and new noise models assign different values to $\lambda_a$, which means $\lambda_a$ is unlearnable. This proof naturally implies that using other noisy gates from the gate set (that are subject to different unknown noise channels) does not change the learnability of Pauli fidelities. More details of the proof are given in Supplementary Section II B. As a side remark, it is known that under the stronger assumption of gate-independent noise (where different multi-qubit gates are assumed to have the same noise channel), the noise channel is fully learnable[28–30].

Theorem 1 provides a simple condition for determining the learnability of individual Pauli fidelities, but it is not sufficient for characterizing the learnability of joint functions of different Pauli fidelities. In the CNOT example, we know that both $\lambda_{IZ}$ and $\lambda_{ZZ}$ are unlearnable, but we also know that their product $\lambda_{IZ}\lambda_{ZZ}$ is learnable. This means that there is only one unlearnable degree of freedom in the two parameters $\{\lambda_{IZ}, \lambda_{ZZ}\}$. In the following we show how to determine

learnable and unlearnable degrees of freedom of Pauli noise, and also generalize the discussion from a single gate to a gate set.

We start by defining learnable information. Consider a Clifford gate set with $m$ gates, where we model each gate as an $n$-qubit gate associated with an $n$-qubit Pauli noise channel. This model is applicable to both individual gates (e.g. a 2-qubit system where each 2-qubit gate is implemented by a different physical process and subject to a different noise channel) as well as parallel applications of gates (e.g. an $n$-qubit system where each "gate" in the gate set is implemented by a layer of 2-qubit gates; the $n$-qubit noise channel models the crosstalk among the 2-qubit gates). The goal is to characterize the learnable degrees of freedom among the $m \cdot 4^n$ parameters.

Recall that the output of cycle benchmarking is a product of Pauli fidelities (including SPAM noise). We further show that without loss of generality this is the only type of information that we need to obtain from quantum experiments for the purpose of noise learning. This is because in general the output probability of any quantum experiment can be expressed as a sum of products of Pauli fidelities, and each individual product can be learned by cycle benchmarking (Supplementary Section IV). We therefore consider learning functions of the noise model that can be expressed as a product of Pauli fidelities (also see below Eq. (7) for a related discussion). This can be reduced to considering functions of the form $f = \sum_{a,\mathcal{G}} v_a^{\mathcal{G}} \cdot l_a^{\mathcal{G}}$, where $l_a^{\mathcal{G}} := \log \lambda_a^{\mathcal{G}}$ is the log Pauli fidelity, $v_a^{\mathcal{G}} \in \mathbb{R}$, and the superscript $\mathcal{G}$ denotes the corresponding Clifford gate. In the CNOT example $l_{IZ} + l_{ZZ}$ is a learnable function. The idea of learning log Pauli fidelities in benchmarking has also been considered in[15,31]. The advantage of considering log Pauli fidelities here is that the set of all learnable functions $f$ forms a vector space. Therefore to characterize all independent learnable degrees of freedom, we only need to determine a basis of the vector space.

Recall that the reason that $l_{IZ} + l_{ZZ}$ is learnable in the CNOT example is because the path of Pauli operator in the cycle benchmarking circuit forms a cycle $IZ \to ZZ \to IZ \to \cdots$, and the product of Pauli fidelities along the cycle ($\lambda_{IZ}\lambda_{ZZ}$) can be learned via curve fitting. In general, as we can also insert single qubit Clifford gates in between, we do not need to differentiate between $X, Y, Z$. We therefore consider the pattern transfer graph associated with a Clifford gate set where vertices corresponds to binary Pauli weight patterns and each edge is labeled by the Pauli fidelity of the incoming Pauli operator. The graph has $2^n$ vertices and $m \cdot 4^n$ directed edges. They can also be merged to form the pattern transfer graph of the gate set {CNOT, SWAP}. Figure 2 shows the pattern transfer graph of CNOT, SWAP, and the gate set of {CNOT, SWAP}. Consider an arbitrary cycle in the pattern transfer graph $C = (e_1, ..., e_k)$ where each edge $e_i$ is associated with some Pauli fidelity $\lambda_i$. Following Fig. 1(b), a cycle benchmarking circuit can be constructed which learns the product of the Pauli fidelites along the cycle, or equivalently the function $f_C := \sum_{e_i \in C} \log \lambda_i$ can be learned. This implies that the set of functions defined by linear combination of cycles $\{\sum_{C \in \text{cycles}} \alpha_C f_C : \alpha_C \in \mathbb{R}\}$ are learnable. In the following we show that this in fact corresponds to all learnable information about Pauli noise.

We label the edges of the pattern transfer graph as $e_1, ..., e_M$ where $M = m \cdot 4^n$ and each edge $e_i$ is a variable that represents some log Pauli fidelity. The goal is to characterize the learnability of linear functions of the edge variables $f = \sum_{i=1}^M v_i e_i, v_i \in \mathbb{R}$. The set of linear functions can be equivalently understood as a vector space of dimension $M$, called the edge space of the graph, where $f$ corresponds to a vector $(v_1, ..., v_M)$ and we think of $e_1, ..., e_M$ as the standard basis. Following the above discussion, the cycle space of the graph is defined as $\text{span}\{\sum_{e \in C} e : C$ is a cycle$\}$, which is a subspace of edge space. We also define another subspace, the cut space, as $\text{span}\{\sum_{e \in C}(-1)^{e \text{ from } V_1 \text{ to } V_2} e : C$ is a cut between a partition of vertices $V_1, V_2\}$. It is known that the edge space is the orthogonal direct sum of cycle space and cut space for any graph[32]. Interestingly, we show that the complementarity

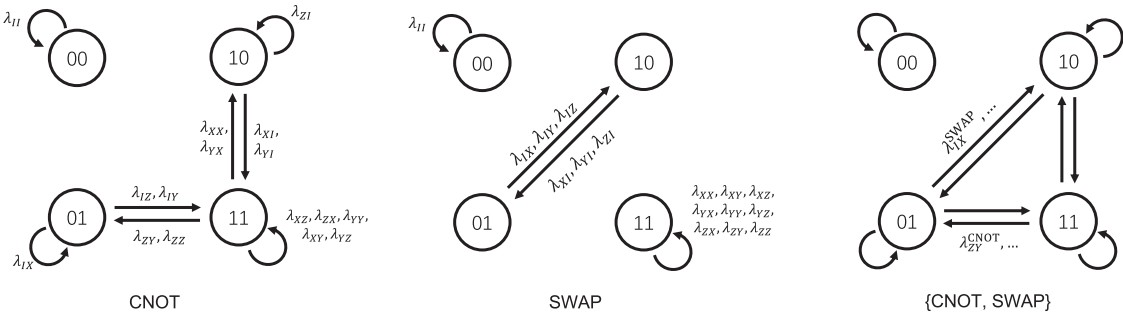

**Fig. 2 | Pattern transfer graph of CNOT, SWAP, and a gate set consisting of CNOT and SWAP.** Here, multiple edges are represented by a single edge with multiple labels. The labels on the first two graphs are gate dependent, though we omit the superscripts of CNOT or SWAP. The labels on the last graph are a combination of the first two graphs and are omitted for clarity.

**Table 1 | A complete basis for the learnable linear functions of log Pauli fidelities and Pauli error rates for a single CNOT/SWAP gate**

| Gate | CNOT | SWAP |
|---|---|---|
| (a) Cycle basis | $l_{II}, l_{ZI}, l_{IX}, l_{ZX}, l_{XZ}, l_{YY}, l_{XY}, l_{YZ},$ <br> $l_{IZ} + l_{ZZ}, l_{IY} + l_{ZY}, l_{IZ} + l_{ZY},$ <br> $l_{XI} + l_{XX}, l_{YI} + l_{YX}, l_{XI} + l_{YX}$ | $l_{II}, l_{XX}, l_{XY}, l_{XZ}, l_{YX}, l_{YY}, l_{YZ}, l_{ZX}, l_{ZY},$ <br> $l_{ZZ}, l_{IX} + l_{XI}, l_{IY} + l_{YI}, l_{IZ} + l_{ZI},$ <br> $l_{XI} + l_{IY}, l_{XI} + l_{IZ}$ |
| (b) Learnable Pauli fidelities | $\lambda_{II}, \lambda_{ZI}, \lambda_{IX}, \lambda_{ZX}, \lambda_{XZ}, \lambda_{YY}, \lambda_{XY}, \lambda_{YZ},$ <br> $\lambda_{IZ} \cdot \lambda_{ZZ}, \lambda_{IY} \cdot \lambda_{ZY}, \lambda_{IZ} \cdot \lambda_{ZY},$ <br> $\lambda_{XI} \cdot \lambda_{XX}, \lambda_{YI} \cdot \lambda_{YX}, \lambda_{XI} \cdot \lambda_{YX}$ | $\lambda_{II}, \lambda_{XX}, \lambda_{XY}, \lambda_{XZ}, \lambda_{YX}, \lambda_{YY}, \lambda_{YZ}, \lambda_{ZX}, \lambda_{ZY},$ <br> $\lambda_{ZZ}, \lambda_{IX} \cdot \lambda_{XI}, \lambda_{IY} \cdot \lambda_{YI}, \lambda_{IZ} \cdot \lambda_{ZI},$ <br> $\lambda_{XI} \cdot \lambda_{IY}, \lambda_{XI} \cdot \lambda_{IZ}$ |
| (c) Learnable Pauli errors | $p_{II}, p_{ZI}, p_{IX}, p_{ZX}, p_{XZ}, p_{YY}, p_{XY}, p_{YZ},$ <br> $p_{IZ} + p_{ZZ}, p_{IY} + p_{ZY}, p_{IZ} + p_{ZY},$ <br> $p_{XI} + p_{XX}, p_{YI} + p_{YX}, p_{XI} + p_{YX}$ | $p_{II}, p_{XX}, p_{XY}, p_{XZ}, p_{YX}, p_{YY}, p_{YZ}, p_{ZX}, p_{ZY},$ <br> $p_{ZZ}, p_{IX} + p_{XI}, p_{IY} + p_{YI}, p_{IZ} + p_{ZI},$ <br> $p_{XI} + p_{IY}, p_{XI} + p_{IZ}$ |
| (d) Unlearnable degrees of freedom | $\lambda_{XI}, \lambda_{IZ}$ | $\lambda_{XI}$ |

between cycle and cut space happens to be the dividing line that determines the learnability of Pauli noise.

**Theorem 2.** The vector space of learnable functions of the Pauli noise channels associated with an $n$-qubit Clifford gate set is equivalent to the cycle space of the pattern transfer graph. In other words,

$$All\ information \equiv Edge\ space,$$
$$Learnable\ information \equiv Cycle\ space,\quad\quad (6)$$
$$Unlearnable\ information \equiv Cut\ space.$$

This implies that the number of unlearnable degrees of freedom equals $2^n - c$, where $c$ is the number of connected components of the pattern transfer graph.

The learnability of cycle space follows from cycle benchmarking as discussed above. To prove the unlearnability of cut space, we use a similar argument as in Theorem 1 and show that a gauge transformation can be constructed for each cut in the pattern transfer graph. By linearity, this implies that any vector in the cut space corresponds to a gauge transformation. By definition, a learnable function must be orthogonal to all such vectors and thus orthogonal to the entire cut space. More details of the proof are given in Supplementary Section II C.

It is a well-known fact in graph theory that the cycle space of a directed graph $G = (V, E)$ has dimension $|E| - |V| + c$ while the cut space has dimension $|V| - c$, where $c \geq 1$ is the number of connected components in $G^{32}$ (a (weakly) connected component is a maximal subgraph in which every vertex is reachable from every other vertex via an undirected path). Theorem 2 implies that among the $m \cdot 4^n$ degrees of freedom of the Pauli noise associated with a Clifford gate set, there are $2^n - c$ unlearnable degrees of freedom. This shows that while the number of unlearnable degrees of freedom can be exponentially large, they only occupy an exponentially small fraction of the entire space. In addition, a cycle and cut basis can be efficiently determined for a given

graph, though in our case this takes exponential time because the pattern transfer graph itself is exponentially large. However, computing the cycle/cut basis is not the bottleneck as the information to be learned also grows exponentially with the number of qubits. For small system sizes such as 2-qubit Clifford gates, we can write down a cycle basis as shown in Table 1(a) for the CNOT and SWAP gates, which represents all learnable information about these gates. The CNOT gate has 2 unlearnable degrees of freedom while the SWAP gate has 1 unlearnable degree of freedom. As the pattern transfer graph has at least 2 connected components, we conclude that the Pauli noise channel of a 2-qubit Clifford gate has at most 2 unlearnable degrees of freedom. Note that when treating {CNOT, SWAP} together as a gate set, there are only 2 unlearnable degrees of freedom according to Theorem 2 instead of $2 + 1 = 3$, because there is one additional learnable degree of freedom (such as $l_{IZ}^{CNOT} + l_{XX}^{CNOT} + l_{XI}^{SWAP}$) that is a joint function of the two gates.

Finally, the learnability of Pauli errors can be determined by the learnability of Pauli fidelities according to the Walsh-Hadamard transform $p_a = \frac{1}{4^n}\sum_{b \in P^n}\lambda_b(-1)^{\langle a,b\rangle}$. An issue here is that Pauli errors are linear functions of $\{\lambda_b\}$ instead of $\{\log\lambda_b\}$. Here we make a standard assumption in the literature[9,10] that the total Pauli error is sufficiently small. In this case all individual Pauli errors are close to 0 while all individual Pauli fidelities are close to 1. Therefore the Pauli errors can be estimated via

$$p_a = \frac{1}{4^n}\sum_{b \in P^n}\lambda_b(-1)^{\langle a,b\rangle} \approx \frac{1}{4^n}\sum_{b \in P^n}(-1)^{\langle a,b\rangle}(1 + \log\lambda_b),\quad (7)$$

which means that their learnability can be determined by Theorem 2. In fact it has been suggested[31] that any function of Pauli fidelities can be estimated in this way (as a linear function of log Pauli fidelities) up to a first-order approximation, which means that the learnability of any function of Pauli fidelities can be determined by Theorem 2. In Table 1 (c) we show the learnable Pauli errors for CNOT and SWAP, where

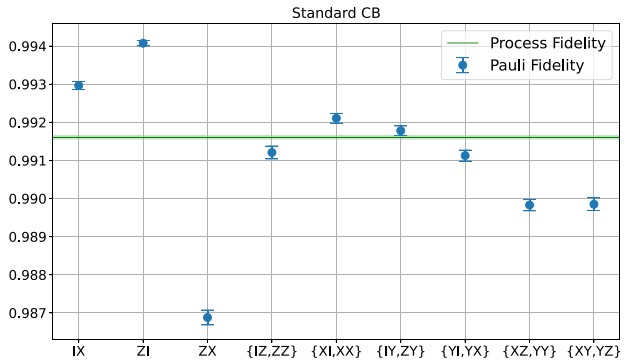

**Fig. 3 | Estimates of Pauli fidelities of IBM's CNOT gate via standard CB (left) and CB with interleaved gates (right), using circuits shown in Fig. 1.** Data are collected from `ibmq_montreal` on 2022-03-23. Each Pauli fidelity is fitted using seven different circuit depths $L = [2, 2^2, \ldots, 2^7]$. For each depth $C = 60$ random circuits and 1000 shots of measurements are used. Throughout this paper, the error bar represents the standard error.

"learnable" is in an approximate sense up to Eq. (7). Interestingly, for these two gates, the learnable functions of Pauli errors have the same form as the cycle basis, i.e. the cycle space is invariant under Walsh-Hadamard transform. We calculate the learnable Pauli errors for up to 4-qubit random Clifford gates and this seems to be true in general. We leave a rigorous investigation into this phenomenon for future work.

### Experiments on IBM Quantum hardware

We demonstrate our theory on IBM quantum hardware[24] using a minimal example – characterizing the noise channel of a CNOT gate. In our experiments both the gate noise and SPAM noise are twirled into Pauli noise using randomized compiling. In the following we show how to extract all learnable information of Pauli noise SPAM-robustly, and also attempt to estimate the unlearnable degrees of freedom by making additional assumptions.

First, we conduct two types of cycle benchmarking (CB) experiments, the standard CB and CB with interleaving single-qubit gates (called interleaved CB), as shown in Fig. 1. The results are shown in Fig. 3. Here a set of two Pauli labels in the $x$-axis (e.g., {$IZ, ZZ$}) corresponds to the geometric mean of the Pauli fidelity (e.g., $\sqrt{\lambda_{IZ}\lambda_{ZZ}}$). Comparing to Table 1, we see that all learnable information of Pauli fidelities (including learnable individual and 2-product) are successfully extracted. Also note from Fig. 3 that the two types of CB experiments give consistent estimates, in terms of both the process fidelity and individual Pauli fidelities (e.g., $\sqrt{\lambda_{XZ}\lambda_{YY}}$ estimated from standard CB is consistent with $\lambda_{XZ}$ and $\lambda_{YY}$ from interleaved CB).

We have shown that all 13 learnable degrees of freedom (excluding the trivial $\lambda_{II} = 1$) are extracted in Fig. 3 by comparing with Table 1, and there remain 2 unlearnable degrees of freedom. We can bound the feasible region of the 2 unlearnable degrees of freedom using physical constraints, i.e., the reconstructed Pauli noise channel must be completely positive. This is equivalent to requiring $p_a \geq 0$ for all Pauli error rates $p_a$. We choose $\lambda_{XX}$ and $\lambda_{ZZ}$ as a representation of the unlearnable degrees of freedom, and plot the calculated feasible region in Fig. 4(a), which happens to be a rectangular area. We also calculate the feasible region for each unlearnable Pauli fidelity and Pauli error rate, which are presented in Fig. 4(b), (c). In particular, we choose two extreme points (blue and green dots in Fig. 4(a)) in the feasible region and plot the corresponding noise model in Fig. 4(b), (c). Note that the (approximately) learnable Pauli error rates (on the left of the red vertical dashed line) are nearly invariant under change of gauge degrees of freedom, but they can be estimated to be negative due to statistical fluctuation. Thus, when we calculate the physical constraints, we only require those unlearnable Pauli error rates (on the right of the red vertical dashed line) to be non-negative.

Next, we explore an approach to estimate the unlearnable information with additional assumptions. Suppose that one can prepare

$|0\rangle^{\otimes n}$ perfectly. Since we assume noiseless single-qubit gates, this means we can prepare a set of perfect tomographically complete states $\{|0/1\rangle, |\pm\rangle, |\pm i\rangle\}$. In this case, all the unlearnable degrees of freedom become learnable, as one can first perform a measurement device tomography, and then directly estimate the process matrix of a noisy gate with measurement error mitigated[25]. Following this general idea, we propose a variant of cycle benchmarking for Pauli noise characterization, which we call intercept CB as it uses the information of intercept in a standard cycle benchmarking protocol. Given an $n$-qubit Clifford gate $\mathcal{G}$, let $m_0$ be the smallest positive integer such that $\mathcal{G}^{m_0} = \mathcal{I}$. For any Pauli fidelity $\lambda_a$ (regardless of whether learnable or not according to Theorem 1), consider the following two CB experiments using the standard circuit as in Fig. 1(a). First, prepare an eigenstate of $P_a$, run CB with depth $lm_0 + 1$ for some non-negative integer $l$, and estimate the expectation value of $P_b := \mathcal{G}(P_a)$. The result equals

$$\mathbb{E}\langle P_b \rangle_{lm_0+1} = \lambda_{P_a}^S \lambda_{P_b}^M \lambda_a \left( \prod_{k=1}^{m_0} \lambda_{\mathcal{G}^k(P_a)} \right)^l, \tag{8}$$

where $\lambda_{P_{a/b}}^{S/M}$ is the Pauli fidelity of the state preparation and measurement noise channel, respectively (earlier we have absorbed these two coefficients into a single coefficient $A$ for simplicity). Second, prepare an eigenstate of $P_b$, run CB with depth $lm_0$, and estimate the expectation value of $P_b$. The result equals

$$\mathbb{E}\langle P_b \rangle_{lm_0} = \lambda_{P_b}^S \lambda_{P_b}^M \left( \prod_{k=1}^{m_0} \lambda_{\mathcal{G}^k(P_a)} \right)^l. \tag{9}$$

By fitting both $\mathbb{E}\langle P_b \rangle_{lm_0+1}$ and $\mathbb{E}\langle P_b \rangle_{lm_0}$ as exponential decays in $l$, extracting the intercepts (function values at $l = 0$), and taking the ratio, we obtain an estimator $\widehat{\lambda}_a^{\text{ICB}}$ that is asymptotically unbiased to $\lambda_a \cdot \lambda_{P_a}^S / \lambda_{P_b}^S$. This estimator is robust against measurement noise. Note that $\lambda_{P_a}^S = \lambda_{P_b}^S = 1$ if we assume perfect initial state preparation, and in this case the above shows that $\lambda_a$ is learnable, and thus the entire Pauli noise channel is learnable. We note that, instead of fitting an exponential decay in $l$, one could in principle just take $l = 0$ and estimate the ratio of $\mathbb{E}\langle P_b \rangle_0$ and $\mathbb{E}\langle P_b \rangle_1$, which also yields a consistent estimate for $\lambda_a \cdot \lambda_{P_a}^S / \lambda_{P_b}^S$. If one has already obtained all the learnable information from previous experiments, this could be a more efficient approach. However, if one has not done those experiments, the intercept CB with multiple depths can estimate the intercept (unlearnable information) and slope (learnable information) simultaneously, which is more sample efficient.

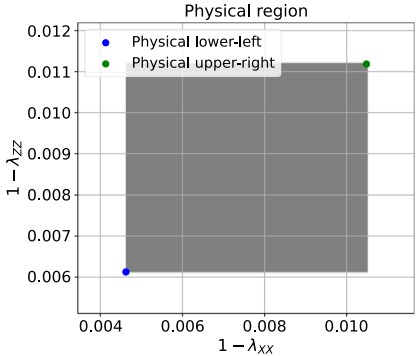

(a) feasible region

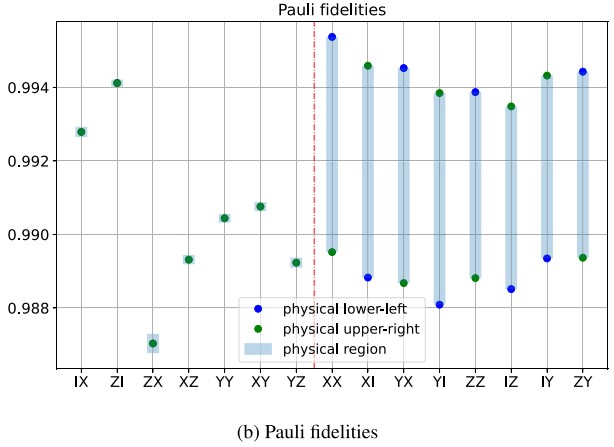

(b) Pauli fidelities

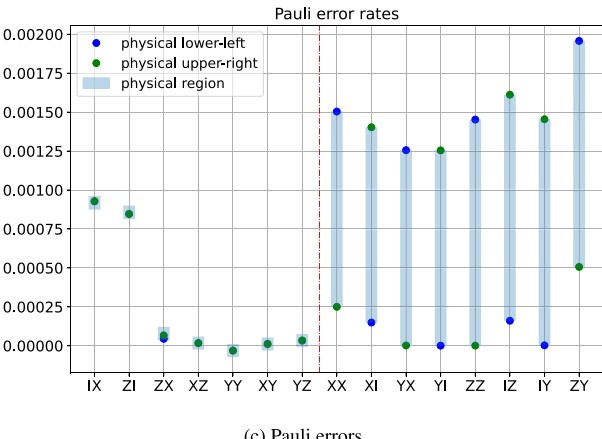

(c) Pauli errors

**Fig. 4 | Feasible region of the learned Pauli noise model, using data from Fig. 3. a** Feasible region of the unlearnable degrees of freedom in terms of $\lambda_{XX}$ and $\lambda_{ZZ}$.

**b** Feasible region of individual Pauli fidelities. **c** Feasible region of individual Pauli errors.

We numerically simulate intercept CB for characterizing the CNOT gate under different state preparation (SP) and measurement (M) noise. As shown in Fig. 5, this method yields relatively precise estimate when there is only measurement noise even if the noise is orders of magnitude stronger than the gate noise, but will have large deviation from the true noise model even under small state preparation noise. We refer the reader to Supplementary Section III for more details about the numerical simulation.

Finally, we experimentally implement intercept CB to estimate $\lambda_{XX}$ and $\lambda_{ZZ}$, which are the two unlearnable degrees of freedom of CNOT, allowing us to determine all the Pauli fidelities and Pauli error rates. One challenge in interpreting the results is that we do not know in general whether the low SP noise assumption holds, therefore it is unclear if the learned results should be trusted. However, for the estimate to be correct, it should at least lie in the physically feasible region we obtained earlier in Fig. 4. In Fig. 6, we present our experimental results of intercept CB. It turns out that certain Pauli fidelities are far away from the physical region by several standard deviations. This gives strong evidence that the low SP noise assumption was not true on the platform we used.

The data collected here can further be used to give a lower bound for the SP noise. Suppose we obtain the physical region of $\lambda_a$ to be $[\widehat{\lambda}_{a,\min}, \widehat{\lambda}_{a,\max}]$. Combining with the expression of intercept CB, we have

$$\widehat{\lambda}_a^{\mathrm{ICB}}/\widehat{\lambda}_{a,\max} \leq \lambda_{P_a}^S/\lambda_{P_b}^S \leq \widehat{\lambda}_a^{\mathrm{ICB}}/\widehat{\lambda}_{a,\min}. \tag{10}$$

Applying this to the data of *IZ* and *ZZ* in Fig. 6(a), we have $\lambda_{IZ}^S/\lambda_{ZZ}^S \leq 0.9879(23)$. If we make a physical assumption that the state

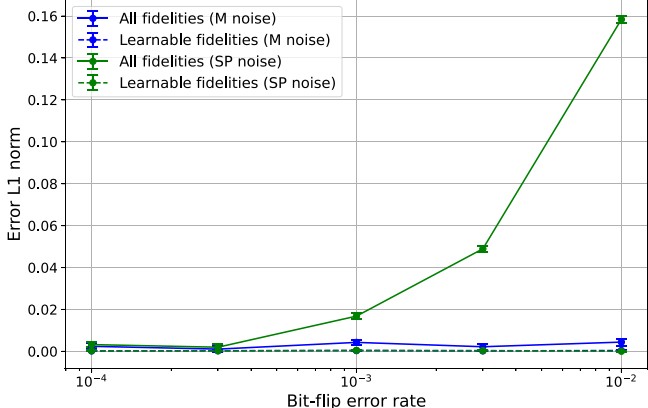

**Fig. 5 | Simulation of intercept CB on CNOT under different SPAM noise rate.** The simulated noise channel is a 2-qubit amplitude damping channel with effective noise rate 5%, and SPAM noise are modeled as bit-flip errors. For the blue (green) lines, we introduce random bit-flip errors to the measurement (state preparation). The solid lines show the $l_1$-distance of the estimated Pauli fidelities from the true Pauli fidelities. The solid lines show the $l_1$-distance of the (individually) learnable Pauli fidelities from the ground truth.

preparation noise is a random bit-flip during the qubit initialization, one can conclude the bit-flip rate on the first qubit is lower bounded by 0.61(12)%. One can in principle bound the bit-flip rate on the second qubit by looking at $\lambda_{XX}^S/\lambda_{XI}^S$. Unfortunately, our estimate of

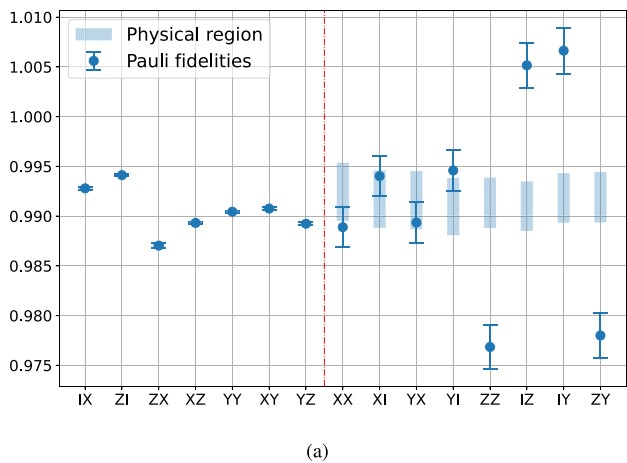

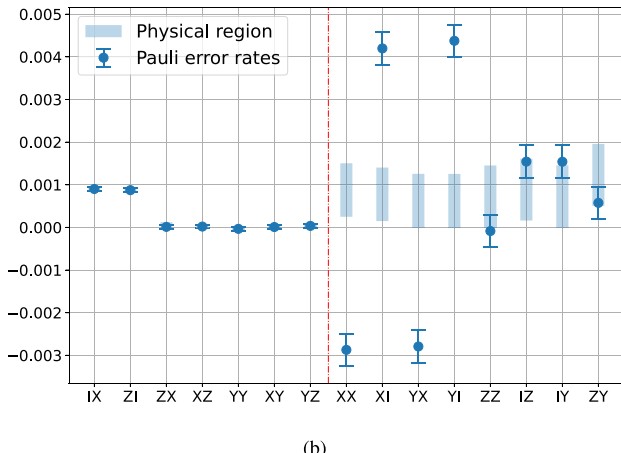

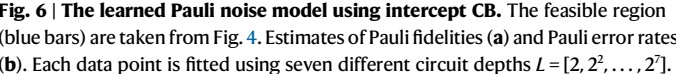

(a)                                              (b)

**Fig. 6 | The learned Pauli noise model using intercept CB.** The feasible region (blue bars) are taken from Fig. 4. Estimates of Pauli fidelities (**a**) and Pauli error rates (**b**). Each data point is fitted using seven different circuit depths $L = [2, 2^2, \ldots, 2^7]$.

For each depth $C = 150$ random circuits and 2000 shots of measurements are used. Data are collected from `ibmq_montreal` on 2022-03-23.

$\lambda_{XX}^S$ from intercept CB falls in the physical region within one standard deviation, so there is no nontrivial lower bound. One could expect obtaining a useful lower bound by looking at a CNOT gate with reversed control and target. The lower bound of SP noise obtained here is completely independent of the measurement noise and does not suffer from the issue of gauge freedom[19], as long as all of our noise assumptions are valid, i.e., there is no significant contribution from time non-stationary, non-Markovian, or single-qubit gate-dependent noise.

## Discussion

We have shown how to characterize the learnability of Pauli noise of Clifford gates and discussed a method to extract unlearnable information by assuming perfect initial state preparation. It is also interesting to consider other physically motivated assumptions on the noise model to avoid unlearnability. For example, we can write down a parameterization of the noise model based on the underlying physical mechanism which may have fewer than $4^n$ parameters. The main issue here is that these assumptions are highly platform-dependent and should be decided case-by-case. Moreover, it is unclear to what extent should the learned results be trusted when additional assumptions are made, since in general we cannot test whether the assumptions hold due to unlearnability.

Another direction to overcome the unlearnability is to change the model of quantum experiments. Here we have been working with the standard model as in gate set tomography, where a quantum measurement decoheres the system and only outputs classical information. However, some platforms might support quantum non-demolition (QND) measurements, and in this case measurements can be applied repeatedly, which could potentially allow more information to be learned[33].

Recently, ref. 30 considered similar issues of noise learnability. They studied a different Pauli noise model with perfect initial state $|0\rangle$, perfect computational basis measurement, and noisy single qubit gates, and showed the existence of unlearnable information. In contrast, here we focus on the learnability of Pauli noise of multi-qubit Clifford gates assuming perfect single-qubit gates (with noisy SPAM), and in practice we make the standard assumption that noise on single-qubit gates is gate-independent (e.g.[23]), in which case our noise learning results are interpreted as characterizing a dressed cycle.

This work leaves open the question of noise learnability for non-Clifford gates. An issue here is that randomized compiling is not known to work with non-Clifford gates in general, so it is unclear if the general CPTP noise learnability problem can be reduced to Pauli noise. Recent

work[14] shows that random quantum circuits can effectively twirl the CPTP noise channel into Pauli noise and can be used to learn the total Pauli error. The question of whether more information can be learned still remains open.

Another issue to address is the scalability in noise learning. It is impossible to estimate all learnable degrees of freedom efficiently as there are exponentially many of them (an exponential lower bound on the sample complexity is shown in[16]). One way to avoid the exponential scaling issue is to assume the noise model has certain special structure (such as sparsity or low-weight) such that the noise model only has polynomially many parameters[10,11,22,34]. It is an interesting open direction to study the characterization of learnability under these assumptions, and we give some related discussions in Supplementary Section II D.

## Data availability

The data generated in this study is available at https://github.com/csenrui/Pauli_Learnability.

## Code availability

The code that supports the findings of this study is available at https://github.com/csenrui/Pauli_Learnability.

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

## Acknowledgements

We thank Ewout van den Berg, Arnaud Carignan-Dugas, Robert Huang, Kristan Temme and Pei Zeng for helpful discussions. S.C. and L.J. acknowledge support from the ARO (W911NF-18-1-0020, W911NF-18-1-0212), ARO MURI (W911NF-16-1-0349, W911NF-21-1-0325), AFOSR MURI (FA9550-19-1-0399, FA9550-21-1-0209), AFRL (FA8649-21-P-0781), DoE Q-NEXT, NSF (OMA-1936118, EEC-1941583, OMA-2137642), NTT Research, and the Packard Foundation (2020-71479). Y.L. was supported by DOE NQISRC QSA grant #FP00010905, Vannevar Bush faculty fellowship N00014-17-1-3025, MURI Grant FA9550-18-1-0161 and NSF award DMR-1747426. A.S. is supported by a Chicago Prize Postdoctoral Fellowship in Theoretical Quantum Science. B.F. acknowledges support from AFOSR (YIP number FA9550-18-1-0148 and FA9550-21-1-0008). This material is based upon work partially supported by the National Science Foundation under Grant CCF-2044923 (CAREER) and by the U.S. Department of Energy, Office of Science, National Quantum Information Science Research Centers. This research used resources of the Oak Ridge Leadership Computing Facility at the Oak Ridge National Laboratory, which is supported by the Office of Science of the U.S. Department of Energy under Contract No. DE-AC05-00OR22725.

## Author contributions

S.C. and Y.L. developed the theory and performed the experiments. B.F. and L.J. supervised the project. S.C., Y.L., M.O., A.S., B.F., and L.J. contributed important ideas during initial discussions and contributed to writing the manuscript.

## Competing interests

The authors declare no competing interests.
