## [Peer Review File · Nature Communications]

The learnability of Pauli noiseREVIEWER COMMENTS

Reviewer #1 (Remarks to the Author):

The authors ask the question, what Pauli fidelities of the noise channel of a Clifford gate are learnable, assuming an unknown Pauli error channel on the state preparation and measurement. They formalize the notion of unlearnability of a given Pauli fidelity as the condition that two error models with differing values of that fidelity can result in identical measurement statistics for all quantum circuits. Their main result is a simple graph-theoretic characterization of the learnable and unlearnable linear functions of the (log) Pauli fidelities. This result is both novel and insightful. It builds upon a large recent body of literature on characterizing Pauli error channels that is largely motivated by the technique of randomized compiling.

The authors also present data from a cycle experiment run on a commercial quantum computer. They compute estimates of the learnable Pauli fidelities and feasible regions of the unlearnable Pauli fidelities. They furthermore show that the unlearnable Pauli fidelities become learnable under the assumption of ideal state preparation. They analyze the data under this additional assumption and find that the Pauli fidelity estimates are inconsistent with the previously obtained feasible regions, demonstrating that this assumption does not hold. It is rare to see model consistency checks of this kind in the quantum computing literature, but they are crucial for determining which assumptions can and cannot be made, and thus which characterization methods are appropriate.

The paper is written in a clear and pedagogical style, and the proofs of the results are correct. I strongly recommend publication. The paper and supplement contain several English grammatical errors, so I recommend a final round of editing for grammar. A list of minor comments follows below.

Main Text

In the last row of Table 1, I was a little confused that p_{XI} , and p_{IZ} are the only “unlearnable Pauli errors” for CNOT, since other Pauli errors individually are also unlearnable, as indicated in Fig. 4 C. Maybe it would be clearer to label this row “unlearnable degrees of freedom” since there are only 2 unlearnable degrees of freedom for CNOT but more than two unlearnable Pauli errors.

Supplemental Information

On page 1, 2nd paragraph, the same notation $pt()$ is used for both the pattern and the weight of a Pauli operator.

On page 9, in lemma 2. Eq. (25), the notation $\dim(Z(G))$ appears twice. It should be $\dim(Z(G)) = \dots$ and $\dim(U(G)) = \dots$, where $Z(G)$ is the cycle space and $U(G)$ is the cut space.

Reviewer #2 (Remarks to the Author):

Summary

The paper studies learnability of Pauli noise channels associated with a gate set of n -qubit Clifford gates. Each gate G is affected by its own noise channel, which is fully characterized by the fidelities of all n -qubit Pauli operators. The concise conclusion of the paper is that the fidelity of Pauli P_a can be learned in a state-preparation and measurement (SPAM) error free manner if and only if $G(P_a)$ and P_a have the same Pauli weight patterns, which are defined as the binary string that indicate the locations of the non-identity terms in the Pauli. By mapping the weight patterns to vertices and defining edges as the transitions between patterns due to the application of a gate, the authors characterize learnability of monomial functions and consequently polynomial functions of Pauli fidelities. By further leveraging graph theoretical results, they fully characterize the learnable and unlearnable degrees of freedom in terms of the dimensions of the cycle and cut spaces, and number of connected components in the graph, in the general case where all noise channels are defined over all n qubits. For two-qubit gates, such as the CNOT gate, the authors show that constraints on Pauli channels can be used to obtain feasible ranges on the Pauli fidelities that cannot be resolved individually, or to obtain lower bounds on the state preparation error. In case of ideal state preparation, it is shown that all Pauli fidelities can be learned.

Comments

The paper is well written and clearly characterizes the learnable Pauli fidelities for noisy Clifford gate sets; a problem that has been an open up to now. My only major question concerns the proposed intercept cycle benchmarking algorithm:

- Is there any advantage to define the expectation values over l repeated cycles of m_0 applications of G ? As an alternative one could simply prepare and measure Pauli $P_b := G(P_a)$ to obtain $\mathbb{E}\langle P_b \rangle_0 = \lambda_{P_b}^S \lambda_{P_b}^M$, and prepare P_a , apply G , and measure P_b to obtain $\mathbb{E}\langle P_b \rangle_1 = \lambda_{P_a}^S \lambda_{P_b}^M \lambda_{P_a}^G$. The ratio $\mathbb{E}\langle P_b \rangle_1 / \mathbb{E}\langle P_b \rangle_0$ is given by $\lambda_{P_a}^G \lambda_{P_a}^S / \lambda_{P_b}^S$. In the case of ideal state preparation we have $P_a^S = P_b^S = 1$, and the above ratio therefore also reduces to the desired quantity $\lambda_{P_a}^G$. Would one of the schemes have a better sample complexity than the other? Is there any advantage in choosing larger l values?

Some minor comments and suggestions are as follows:

- p.3, l.1: Writing that $\{p_a\}$ is a “ 4^n dimensional” probability distribution makes it sound like the distribution is a function of 4^n variables. Perhaps omit “ 4^n dimensional”, the next line already notes that the distribution has $4^n - 1$ degrees of freedom.
- p.3, l.14: “equals $\lambda_{IX}^3 \cdot XI$ up to a \pm sign”; where does the \pm sign come from. If the gate maps IX to $-IX$, would this not be reflected in a negative λ_{IX} in Eq. 2?
- p.3, l.15: “and average over random Pauli sequences”; perhaps clarify that these are the random Pauli twirl sequences. For the expectation to be of the form given by Eq. 3, it may be necessary to twirl the readout as well.
- p.3, l.-5: Strictly speaking, the single-qubit Clifford gates ($\sqrt{Z}\sqrt{X}$) maps XZ to $-YY$ and vice versa.
- p.4,l.13: “single qubit rotation trick inapplicable”, perhaps replace “trick” by “correction” or “tool”.
- p.4, l.-15: since there could be multiple, replace “there exists one bit” by “there exists a bit”
- p.5, Fig. 2: To avoid confusion it would help to mention that the λ values are gate specific. That is, the λ_{IX} term in the CNOT graph may not be the same as the λ_{IX} term in the SWAP graph.
- p.6, l.-14: replace “a set of cycle and cut basis” by “a cycle and cut basis” or, if needed, by “a set of cycle and cut bases”.
- p.7, l.11: “Interestingly, here the learnable functions of Pauli errors have the same form as the cycle basis, *i.e.* the cycle space is invariant under Walsh-Hadamard transform. We leave the proof of this observation for future work”. This sentence makes it sound like this property holds more generally. If it only applies to CNOT and SWAP for now, replace “here” following “Interestingly” by “for these two gates”.
- p.7,l.-4: “and there **remain** 2 unlearnable degrees of freedom”

In contrast to the main paper, the supplement seems to be written in a hurry and needs to be carefully revised. Below are some of the many (minor) issues:

- It is completely beyond me why the authors would use $pt(P_a)$ to denote both the pattern of the Pauli as well as the Hamming weight of the Pauli. Why not simply define pt for the pattern and pw or w for the weight? Note that the current pt cannot be used interchangeably. For instance, in Theorem 1 using the pt interpretation of weight is wrong: For $P_a = IX$ and $\mathcal{G}(P_a) = XI$ we have $pt(\mathcal{G}(P_a)) \neq pt(P_a)$ for the pattern but $pt(\mathcal{G}(P_a)) \neq pt(P_a)$ for the weight. Please rename at least one of the two functions.
- p.1, l.-2: rephrase “ $p := \{p_a\}_a$ is called the Pauli error rates”

- p.2, l.-11: rephrase “We discuss in the main text about how to”
- p.3, l.2: rephrase “in many existing research”
- p.3, l.5: “A Clifford gate ... are viewed as different gates”
- p.3, l.13: “is determined the Pauli channels”
- p.3, l.19: “can be describe by”
- Etc. for the remainder of the supplement . . .
- p.5, l.1: Suggest: “Recall that $\lambda_a^{S/M}$ is the fidelity of the SPAM noise channel for Pauli P_a .”
- p.8ff, “maximal” \rightarrow “maximum” and “minimal” \rightarrow “minimum”
- p.8, Given that $|E|$ is used in Lemma 2, it might help to define $|\Lambda| = |E| \equiv |\mathfrak{G}| \cdot 4^n$, or just use $|E|$ throughout instead of $|\Lambda|$.
- p.9, l.3: Rephrase: “the set of all edges $e = (u, v)$ such that u, v each belongs to one of V_1, V_2 ”. The way it is written now allows us to choose $u \in V_1$ and $v \in V_1$.
- p.9, Eq. 25: Both expressions are for $\dim(Z(G))$, the second probably should be $\dim(U(G))$, otherwise $\dim(Z(G))$ would reduce to $|E|/2$.
- p.12, l.-7: Carefully rephrase “holds either $pt(P) = pt(Q)$ or $pt(P) \neq pt(Q)$ (in the latter case, both sides equals zero)”. Maybe write that “holds when $pt(P) = pt(Q)$, but also for $pt(P) \neq pt(Q)$, in which case both traces are zero.”

Response to referee comments for NCOMMS-22-26755

Senrui Chen, Yunchao Liu, Matthew Otten, Alireza Seif, Bill Fefferman, and Liang Jiang

We greatly appreciate both referees' efforts in reviewing our manuscript and offering many constructive suggestions and comments. Overall, both referees agree our manuscript is well-written and makes meaningful contributions to the field. In the following, we present a point-to-point response to the referees' comments. We also attach a list of additional changes at the end of this document. We hope these revisions address the questions raised by the reviewers.

I. RESPONSE TO REVIEWER #1

The authors ask the question, what Pauli fidelities of the noise channel of a Clifford gate are learnable, assuming an unknown Pauli error channel on the state preparation and measurement. They formalize the notion of unlearnability of a given Pauli fidelity as the condition that two error models with differing values of that fidelity can result in identical measurement statistics for all quantum circuits. Their main result is a simple graph-theoretic characterization of the learnable and unlearnable linear functions of the (log) Pauli fidelities. This result is both novel and insightful. It builds upon a large recent body of literature on characterizing Pauli error channels that is largely motivated by the technique of randomized compiling.

The authors also present data from a cycle experiment run on a commercial quantum computer. They compute estimates of the learnable Pauli fidelities and feasible regions of the unlearnable Pauli fidelities. They furthermore show that the unlearnable Pauli fidelities become learnable under the assumption of ideal state preparation. They analyze the data under this additional assumption and find that the Pauli fidelity estimates are inconsistent with the previously obtained feasible regions, demonstrating that this assumption does not hold. It is rare to see model consistency checks of this kind in the quantum computing literature, but they are crucial for determining which assumptions can and cannot be made, and thus which characterization methods are appropriate.

The paper is written in a clear and pedagogical style, and the proofs of the results are correct. I strongly recommend publication. The paper and supplement contain several English grammatical errors, so I recommend a final round of editing for grammar. A list of minor comments follows below.

Response: We thank the reviewer for the clear summary of our work and all the helpful comments. We have addressed the list of minor comments in below and done another round of grammatical checks.

(Main Text) In the last row of Table 1, I was a little confused that p_{XI} , and p_{IZ} are the only “unlearnable Pauli errors” for CNOT, since other Pauli errors individually are also unlearnable, as indicated in Fig. 4 C. Maybe it would be clearer to label this row “unlearnable degrees of freedom” since there are only 2 unlearnable degrees of freedom for CNOT but more than two unlearnable Pauli errors.

Response: We agree with the referee on this point. We have relabeled the last row of Table 1 as “unlearnable degrees of freedom”. To make things clearer, we add an additional row of “learnable Pauli fidelities” and use Pauli fidelities $(\lambda_{XI}, \lambda_{IZ})$ instead to represent the unlearnable degrees of freedoms.

(Supplemental Information) On page 1, 2nd paragraph, the same notation $\text{pt}()$ is used for both the pattern and the weight of a Pauli operator.

Response: We have removed the definition of Pauli *weights* since it is not used anywhere in the manuscript. Now $\text{pt}(\cdot)$ refers exclusively to Pauli *patterns*.

On page 9, in lemma 2. Eq. (25), the notation $\dim(\mathbb{Z}(G))$ appears twice. It should be $\dim(\mathbb{Z}(G)) = \dots$ and $\dim(\mathbb{U}(G)) = \dots$, where $\mathbb{Z}(G)$ is the cycle space and $\mathbb{U}(G)$ is the cut space.

Response: We have corrected the second $\mathbb{Z}(G)$ to $\mathbb{U}(G)$.

II. RESPONSE TO REVIEWER #2

(Summary) The paper studies learnability of Pauli noise channels associated with a gate set of n -qubit Clifford gates. Each gate G is affected by its own noise channel, which is fully characterized by the fidelities of all n -qubit Pauli operators. The concise conclusion of the paper is that the fidelity of Pauli P_a can be learned in a state-preparation and measurement (SPAM) error free manner if and only if $G(P_a)$ and P_a have the same Pauli weight patterns, which are defined as the binary string that indicate the locations of the non-identity terms in the Pauli. By mapping the weight patterns to vertices and defining edges as the transitions between patterns due to the application of a gate, the authors characterize learnability of monomial functions and consequently polynomial functions of Pauli fidelities. By further leveraging graph theoretical results, they fully characterize the learnable and unlearnable degrees of freedom in terms of the dimensions of the cycle and cut spaces, and number of connected components in the graph, in the general case where all noise channels are defined over all n qubits. For two-qubit gates, such as the CNOT gate, the authors show that constraints on Pauli channels can be used to obtain feasible ranges on the Pauli fidelities that cannot be resolved individually, or to obtain lower bounds on the state preparation error. In case of ideal state preparation, it is shown that all Pauli fidelities can be learned.

Response: We thank the referee for the clear and concise summary of our work.

(Comments) The paper is well written and clearly characterizes the learnable Pauli fidelities for noisy Clifford gate sets; a problem that has been an open up to now. My only major question concerns the proposed intercept cycle benchmarking algorithm: Is there any advantage to define the expectation values over l repeated cycles of m_0 applications of G ? As an alternative one could simply prepare and measure Pauli $P_b := G(P_a)$ to obtain $\mathbb{E}\langle P_b \rangle_0 = \lambda_{P_b}^S \lambda_{P_b}^M$, and prepare P_a , apply G , and measure P_b to obtain $\mathbb{E}\langle P_b \rangle_1 = \lambda_{P_a}^S \lambda_{P_b}^M \lambda_{P_a}^G$. The ratio $\mathbb{E}\langle P_b \rangle_1 / \mathbb{E}\langle P_b \rangle_0$ is given by $\lambda_{P_a}^G \lambda_{P_a}^S / \lambda_{P_b}^S$. In the case of ideal state preparation we have $P_a^S = P_b^S = 1$, and the above ratio therefore also reduces to the desired quantity $\lambda_{P_a}^G$. Would one of the schemes have a better sample complexity than the other? Is there any advantage in choosing larger l values?

FIG. 1. Numerical simulation comparing intercept CB using 6 different depths with the 1-depth method suggested by the referee. The task is to estimate $\lambda_{ZZ}^{\text{CNOT}}$. We set the state preparation to be perfect, so both methods gives consistent estimate for $\lambda_{ZZ}^{\text{CNOT}}$, thus we only compare the standard deviation of the estimator. The circuit depths used by intercept CB is [1, 2, 4, 8, 16, 32]. The total number of random circuits used for both methods are the same. Each random circuits are measured with 2000 shots.

Response:

We thank the referee for raising this interesting question. Indeed, if we are only interested in extracting the intercept information (*i.e.*, $\mathbb{E}\langle P_b \rangle_0$ and $\mathbb{E}\langle P_b \rangle_1$), there should be no advantages in sample complexity for choosing larger l values and fitting the intercept compared to directly measuring the intercept with a “depth-1” circuit (intuitively, the latter should have a better performance). However, intercept CB allows one to simultaneously extract the intercept and slope, which might give us more information from a single experiment. In Fig. 1, we report a numerical simulation comparing intercept CB with 6 different circuit depths and the 1-depth methods suggested by the referee. The two experiments uses the same total number of random circuits N_C . That is, the 1-depth method uses all N_C random circuits to estimate $\mathbb{E}\langle P_b \rangle_0$ and $\mathbb{E}\langle P_b \rangle_1$ while the 6-depth intercept CB method uses $N_C/6$ random circuits for each depth. From the result, we see the two methods have similar standard deviation for their estimation. However, intercept CB also gives estimate for the slope, which represent the geometric mean of some Pauli fidelities. From a theoretical view point, as long as one does not make the circuit too deep and decay too severe, estimating the intercept from multiple depths should not have a sample complexity much worse than the 1-depth method (thinking about the extreme case no decay happens at all. In that case, estimating the intercept at different circuit depths has no difference). Therefore, we think it would not hurt to use intercept CB which always yields more information.

We do agree that it could be more efficient to just do a 1-depth experiment if one has already obtained the slope information from other experiments. In the main text, we have added the following discussion: *We note that, instead of fitting an exponential decay in l , one could in principle just take $l = 0$ and estimate the ratio of $\mathbb{E}\langle P_b \rangle_0$ and $\mathbb{E}\langle P_b \rangle_1$, which also yields a consistent estimate for $\lambda_a \cdot \lambda_{P_a}^S / \lambda_{P_b}^S$. If one has already obtained all the learnable information from previous experiments,*

this could be a more efficient approach. However, if one has not done those experiments, the intercept CB with multiple depths can estimate the intercept (unlearnable information) and slope (learnable information) simultaneously, which is more sample efficient.

We hope the above explanation and modification have addressed the referee's concern. We have thanked the referee for this suggestion in our acknowledgement.

Some minor comments and suggestions are as follows:

p.3, l.1: Writing that $\{p_a\}$ is a "4ⁿ dimensional" probability distribution makes it sound like the distribution is a function of 4ⁿ variables. Perhaps omit "4ⁿ dimensional", the next line already notes that the distribution has 4ⁿ - 1 degrees of freedom.

Response: We thank the referee for all the helpful suggestions and comments hereafter. We agree with referee and have omitted the "4ⁿ dimensional".

p.3, l.14: "equals $\lambda_{IX}^3 \cdot XI$ up to a \pm sign"; where does the \pm sign come from. If the gate maps IX to $-IX$, would this not be reflected in a negative λ_{IX} in Eq. 2?

Response: Here, the sign comes from the layers of random Pauli gates (*e.g.*, $ZXZ = -X$). Since this is a property of Pauli gates and has nothing to do with the noise of CNOT, it should not be reflected in λ_{IX} in Eq. (2). We just need to account for the sign during post-processing in order to obtain a correct estimate of λ_{IX} . We've modified the sentence to make things clearer: ... *up to a \pm sign (which comes from the random Pauli gates and can always be accounted for during post-processing).*

p.3, l.15: "and average over random Pauli sequences"; perhaps clarify that these are the random Pauli twirl sequences. For the expectation to be of the form given by Eq. 3, it may be necessary to twirl the readout as well.

Response: We have added the word "twirling" in order to make the statement clearer. Regarding the second point raised by the referee, we would like to point out that the circuits we used in FIG. 1 already effectively twirl the readout (since there is an layer of random Pauli gates right before the readout). Actually, it is rigorously proved that this type of circuits yield expectation value of the form given by Eq. (3), according to Theorem 1 in the Supplementary Information of Ref. [1]. We now explicitly cite this theorem in the main text.

p.3, l.5: Strictly speaking, the single-qubit Clifford gates ($\sqrt{Z}\sqrt{X}$) maps XZ to $-YY$ and vice versa.

Response: We agree with the referee on this. However, based on the same logic as above, this minus sign has nothing to do with the noise channel associated with CNOT, thus should not be reflected in either λ_{XZ} or λ_{YY} . We would account for the sign (as well as the sign induced by random Pauli gates) during post-processing to ensure correct estimates. We modify the sentence as follows: ... $\sqrt{Z} \otimes \sqrt{X}$ that also maps XZ to YY and vice versa (*up to a minus sign that can always be accounted for during post-processing*).

p.4, l.13: "single qubit rotation trick inapplicable", perhaps replace "trick" by "correction" or "tool".

Response: We have changed the "trick" to "tool".

p.4, l.-15: since there could be multiple, replace "there exists one bit" by "there exists a bit"

Response: We agree and have done the replacement.

p.5, Fig. 2: To avoid confusion it would help to mention that the λ values are gate specific. That is, the λ_{IX} term in the CNOT graph may not be the same as the λ_{IX} term in the SWAP graph.

Response: We find it too crowded to add superscripts of “CNOT” or “SWAP” to all λ , so we only do this for two specific λ in the right-most graph of Fig. 2 to show the idea. We also now explicitly mention in the caption that the λ values are gate specific. We hope this is sufficient to clear any confusion.

p.6, l.-14: replace "a set of cycle and cut basis" by "a cycle and cut basis" or, if needed, by "a set of cycle and cut bases".

Response: We agree “a cycle and cut basis” is a more precise statement and have changed.

p.7, l.11: "Interestingly, here the learnable functions of Pauli errors have the same form as the cycle basis, i.e. the cycle space is invariant under Walsh-Hadamard transform. We leave the proof of this observation for future work". This sentence makes it sound like this property holds more generally. If it only applies to CNOT and SWAP for now, replace "here" following "Interestingly" by "for these two gates".

Response: We agree that this is only rigorously true for CNOT and SWAP and have changed “here” to “for these two gates” as the referee suggested. However, we have tried other Clifford gates and find this seems to be a generic phenomenon. We added the following sentences to the main text: *We calculate the learnable Pauli errors for up to 4-qubit random Clifford gates and this seems to be true in general. We leave a rigorous investigation into this phenomenon for future work.*

p.7,l.-4: "and there remain 2 unlearnable degrees of freedom"

Response: We agree and have corrected the typo.

In contrast to the main paper, the supplement seems to be written in a hurry and needs to be carefully revised. Below are some of the many (minor) issues:

It is completely beyond me why the authors would use $pt(P_a)$ to denote both the pattern of the Pauli as well as the Hamming weight of the Pauli. Why not simply define pt for the pattern and pw or w for the weight? Note that the current pt cannot be used interchangeably. For instance, in Theorem 1 using the pt interpretation of weight is wrong: For $P_a = IX$ and $\mathcal{G}(P_a) = XI$ we have $pt(\mathcal{G}(P_a)) \neq pt(P_a)$ for the pattern but $pt(\mathcal{G}(P_a)) = pt(P_a)$ for the weight. Please rename at least one of the two functions.

Response: We apologize for the confusion. This is a typo. Since the definition of Pauli *weight* is actually not used anywhere in the manuscript, we have removed it. Now $pt(\cdot)$ refers exclusively to Pauli *patterns* and there should be no more ambiguity.

p.1, l.-2: rephrase " $p := \{p_a\}_a$ is called the Pauli error rates"

Response: We change “is” to “are”. Similarly for the definition of Pauli fidelities after this.

p.2, l.-11: rephrase "We discuss in the main text about how to"

Response: We have rephrased this sentence as “*As shown in the main text, the learnable information about Pauli noise can be extracted in a much more practical setting using cycle benchmarking [1] and its variant.*”

p.3, l.2: rephrase "in many existing research"

Response: We have rephrased this sentence as “*Such approximation is widely adopted in the literature [1, 2] with slight modifications.*”

p.3, l.5: "A Clifford gate ... are viewed as different gates"

Response: For grammatical correctness, we changed the sentence to “*A Clifford gate acting on a different (ordered) subset of qubits is viewed as a different gate and can thus have a different noise channel.*” To make things clearer, we add an example: “*(e.g., $CNOT_{12}$, $CNOT_{21}$, $CNOT_{23}$ have different noise channels.)*”

p.3, l.13: "is determined the Pauli channels"

Response: We have corrected the typo: “... is determined *by* the Pauli channels ...”

p.3, l.19: "can be describe by"

Response: We have corrected the typo: “describe” \mapsto “described”.

Etc. for the remainder of the supplement ...

Response: We have ran another round of grammatical check for the supplemental material and corrected several typos.

p.5, l.1: Suggest: "Recall that $\lambda_a^{S/M}$ is the fidelity of the SPAM noise channel for Pauli P_a ."

Response: We have changed as suggested.

p.8ff, "maximal" \rightarrow "maximum" and "minimal" \rightarrow "minimum"

Response: We have changed as suggested.

p.8, Given that $|E|$ is used in Lemma 2, it might help to define $|\Lambda| = |E| \equiv |\mathfrak{G}| \cdot 4^n$, or just use $|E|$ throughout instead of $|\Lambda|$.

Response: We prefer keeping both $|E|$ and $|\Lambda|$ since they have different meanings - the number of edges and the number of Pauli fidelities with gate noise. We add $|E| = |\Lambda| = |\mathfrak{G}| \cdot 4^n$ as suggested right after Definition 3.

p.9, l.3: Rephrase: "the set of all edges $e = (u, v)$ such that u, v each belongs to one of V_1, V_2 ". The way it is written now allows us to choose $u \in V_1$ and $v \in V_1$.

Response: We change this sentence to “*the set of all edges $e = (u, v)$ such that one of u, v belongs to V_1 and the other belongs to V_2 .*”

p.9, Eq. 25: Both expressions are for $\dim(Z(G))$, the second probably should be $\dim(U(G))$, otherwise $\dim(Z(G))$ would reduce to $|E|/2$.

Response: We agree and have corrected the second $Z(G)$ to $U(G)$.

p.12, 1.-7: Carefully rephrase "holds either $pt(P) = pt(Q)$ or $pt(P) \neq pt(Q)$ (in the latter case, both sides equals zero)". Maybe write that "holds when $pt(P) = pt(Q)$, but also for $pt(P) \neq pt(Q)$, in which case both traces are zero."

Response: We re-organize this paragraph as below: *We see that $\text{Tr}(P \cdot \mathcal{U}'(Q)) = \text{Tr}(P \cdot \mathcal{U}(Q))$ if $pt(P) = pt(Q)$. A crucial observation is that a product of single-qubit unitaries can never change the pattern of the input Pauli. More precisely, $\mathcal{U}(Q)$ is a linear combination of Pauli operators with the same pattern as Q . Therefore, if $pt(P) \neq pt(Q)$, we would have $\text{Tr}(P \cdot \mathcal{U}'(Q)) = \text{Tr}(P \cdot \mathcal{U}(Q)) = 0$. Combining the two cases, we conclude $\mathcal{U}' = \mathcal{U}$, i.e., the single-qubit unitaries are still noiseless in \mathcal{N}' .*

III. LIST OF ADDITIONAL CHANGES

1. We rescaled the font size of some figures to improve the visual effects.
2. According to Nature Communications' data presentation policy, we add the following disclaimer to the caption of Fig. 3: *Throughout this paper, the error bar represents the standard error.*
3. We corrected some typos and made some minor modifications (see the diff document for details).

-
- [1] A. Erhard, J. J. Wallman, L. Postler, M. Meth, R. Stricker, E. A. Martinez, P. Schindler, T. Monz, J. Emerson, and R. Blatt, Characterizing large-scale quantum computers via cycle benchmarking, Nature Communications **10**, 5347 (2019).
- [2] J. J. Wallman and J. Emerson, Noise tailoring for scalable quantum computation via randomized compiling, Physical Review A **94**, 052325 (2016).

REVIEWERS' COMMENTS

Reviewer #2 (Remarks to the Author):

The authors have thoroughly addressed all my previous comments and carefully revised the manuscript. The manuscript looks very good now and I have no additional comments to make.